# The Influence of Calcium Ions on the Electrotransfer Efficiency of Plasmid DNA and Cell Viability

Rūta Palepšienė, Martynas Maciulevičius [ID], Paulius Ruzgys [ID], Baltramiejus Jakštys [ID] and Saulius Šatkauskas *[ID]

Biophysical Research Group, Faculty of Natural Sciences, Vytautas Magnus University, Vileikos St. 8, LT-44404 Kaunas, Lithuania
* Correspondence: saulius.satkauskas@vdu.lt

**Abstract:** Gene electrotransfer (GET) is recognized as a promising technique for the development of an efficient tool for gene therapy. Such a therapy would have applications in the treatment of a variety of genetic diseases, including cancer. However, despite its wide applicability, the technique is limited by the lack of understanding of the fundamental mechanism of electroporation as well as other important factors that directly or indirectly influence its success rate. In the current study, we analyzed the impact of low concentrations (0–1 mM) of $Ca^{2+}$ on the process of DNA electrotransfer using flow cytometry. The results revealed that the presence of a $CaCl_2$ concentration as low as 0.25 mM decreased the efficiency of GET by ~1.5-fold and cell viability decreased by ~2–3-fold. In addition, we determined that the observed phenomenon of the decrease in pDNA electrotransfer due to the influence of $Ca^{2+}$ was not the consequence of cell death but rather should be attributed to secondary mechanisms. The data presented in this study provide an insight into the importance of $Ca^{2+}$ in the process of gene electrotransfer that may be directly applicable to in vivo settings.

**Keywords:** calcium; electroporation; gene electrotransfer; cell death; electrotransfection

## 1. Introduction

Electroporation is a widely used biophysical technique that is exploited with the aim to significantly increase cell membrane permeability to exogenous biomolecules by applying external electric fields [1]. The method is routinely used for the transfer of small molecules, chemotherapeutic agents, and nucleic acids (RNA or DNA) [2–4]. The ability to effectively electroporate different types of cells (mammalian, plant, and microorganism) and transfer different types of molecules ensures the wide applicability of the method in biotechnology, clinics, and food processing [5–7].

The electrotransfer of DNA, also known as cell electrotransfection, is a commonly exploited technique for both in vitro and in vivo applications. Even though the first experiments in DNA electrotransfer were presented more than 30 years ago, the exact mechanism of intracellular and intranuclear delivery is still under investigation [1,8,9]. To date, research has revealed the interdependency between DNA electrotransfer and electrophoretic forces: if the critical value of the external electric field is reached, DNA interacts with the plasma membrane exclusively at the cathode side. DNA is then translocated across the plasma membrane and transported to the nucleus for subsequent gene expression [5,8,10–12]. Even though pDNA electrotransfer in vitro might serve as a relatively effective technique with the selection of certain electroporation parameters and cell lines, there is much that is still unknown [7,13]. Therefore, an understanding of the fundamental mechanisms of DNA electrotransfer—as well as knowledge regarding other important key factors that influence the efficiency of electrotransfer—is crucial and must be taken into account.

$Ca^{2+}$ is one of the most important secondary messengers and is involved in a variety of different cellular processes. Therefore, the concentration of calcium ions in the cell is tightly regulated and can be significantly affected as a response to the influx of extracellular $Ca^{2+}$

after electroporation [14–16]. Changes in the number of intracellular $Ca^{2+}$ have been proved to be associated with ATP depletion, ROS generation, and the activation of lipases and proteases that lead to apoptotic and/or necrotic cell death [17]. It has also been suggested that $Ca^{2+}$ is included in the process of pDNA electrotransfection [18]. It was proposed that bivalent cations might enhance the interaction between pDNA and membrane during the application of electric pulses. However, $Ca^{2+}$ can also bind to polyanionic pDNA, neutralize its charge, and cause aggregation, which therefore leads to decreased transfection [18]. Thereby, the importance of the bivalent cations in the process of electrotransfection is still ambiguous.

To date, only a few studies have analyzed the influence of $Ca^{2+}$ in the mechanism of DNA electrotransfer; these mainly focused on higher (>1 mM) $CaCl_2$ concentrations [19,20]. In this study, we investigated the influence of low concentrations (0–1 mM) of $Ca^{2+}$ on DNA electrotransfer efficiency. Our research demonstrated the significant impact of small amounts of $Ca^{2+}$ on the efficiency of DNA electrotransfer and cell viability in a concentration-dependent manner. In addition, we explicitly demonstrated that $Ca^{2+}$ in a range of 0–1 mM concentrations and co-administered with pDNA can be exploited as a tool for the control of the efficiency of DNA transfection

## 2. Materials and Methods

### 2.1. Cell Culture

The experiments were performed using a Chinese hamster ovary (CHO-K1) cell line (European Collection of Authenticated Cell Cultures; 85050302). Cells were maintained in Dulbecco's Modified Eagle Medium (DMEM) (Sigma-Aldrich, St. Louis, MO, USA) supplemented with 10% fetal bovine serum (FBS) (Gibco™, Gaithersburg, MD, USA), 1% l-glutamine (Sigma-Aldrich, St. Louis, MO, USA), and 1% penicillin–streptomycin solution (Sigma-Aldrich, St. Louis, MO, USA) in a humidified incubator at 37 °C and 5% $CO_2$. The cells were passaged every 2–3 days and a day before the experiment.

### 2.2. Plasmid DNA Preparation

A pEGFP-N1 plasmid (4.7 kb) (Lonza, Walkersville, MD, USA) that encoded for the enhanced green fluorescent protein (GFP) was used in the experiments. The plasmid purification was performed using the Plasmid Giga Kit (Qiagen, Hilden, Germany) according to the guidelines provided by the manufacturer. The nucleic acid purity and concentration were evaluated using a spectrophotometer (Nanodrop 2000, Thermo Fisher Scientific, Washington, DC, USA).

### 2.3. Cell Harvesting and Electroporation

For the electroporation experiments, the cells were harvested according to the standard procedure. Briefly, the growth medium was removed, the cells were washed with $1\times$ PBS (Sigma-Aldrich, St. Louis, MO, USA), and then the cells were incubated with Trypsin-EDTA solution (Sigma-Aldrich, St. Louis, MO, USA) at 37 °C for 4 min. The cells were resuspended in an equal volume of medium, centrifuged for 2 min at 1000 rpm, and counted using a hemocytometer (Paul Marienfeld GmbH & Co. KG, Lauda-Königshofen, Germany). Then, the required number of cells was mixed with a HEPES-based electroporation medium containing 10 mM HEPES (Lonza, Basel, Switzerland), 250 mM sucrose (Sigma-Aldrich, St. Louis, MO, USA), and 1 mM $MgCl_2$ (Sigma-Aldrich, St. Louis, MO, USA). The medium was prepared in Milli-Q water and then supplemented with different concentrations (0, 0.01, 0.1, 0.25, 0.5, or 1 mM) of $CaCl_2$ (Sigma-Aldrich, St. Louis, MO, USA). The concentration of cells in the sample was kept constant at $2 \times 10^6$ cells/mL. All of the experiments were performed using 2 sets of electroporation parameters: 1400 V/cm for 100 μs and 1200 V/cm for 250 μs using a single square-wave electric pulse. The electrodes were composed of stainless steel with a 2 mm gap in between them. An electroporator (AmberCharge, Palanga, Lithuania) was used for the generation of electric pulses. The principal scheme of the study is provided in Figure 1.

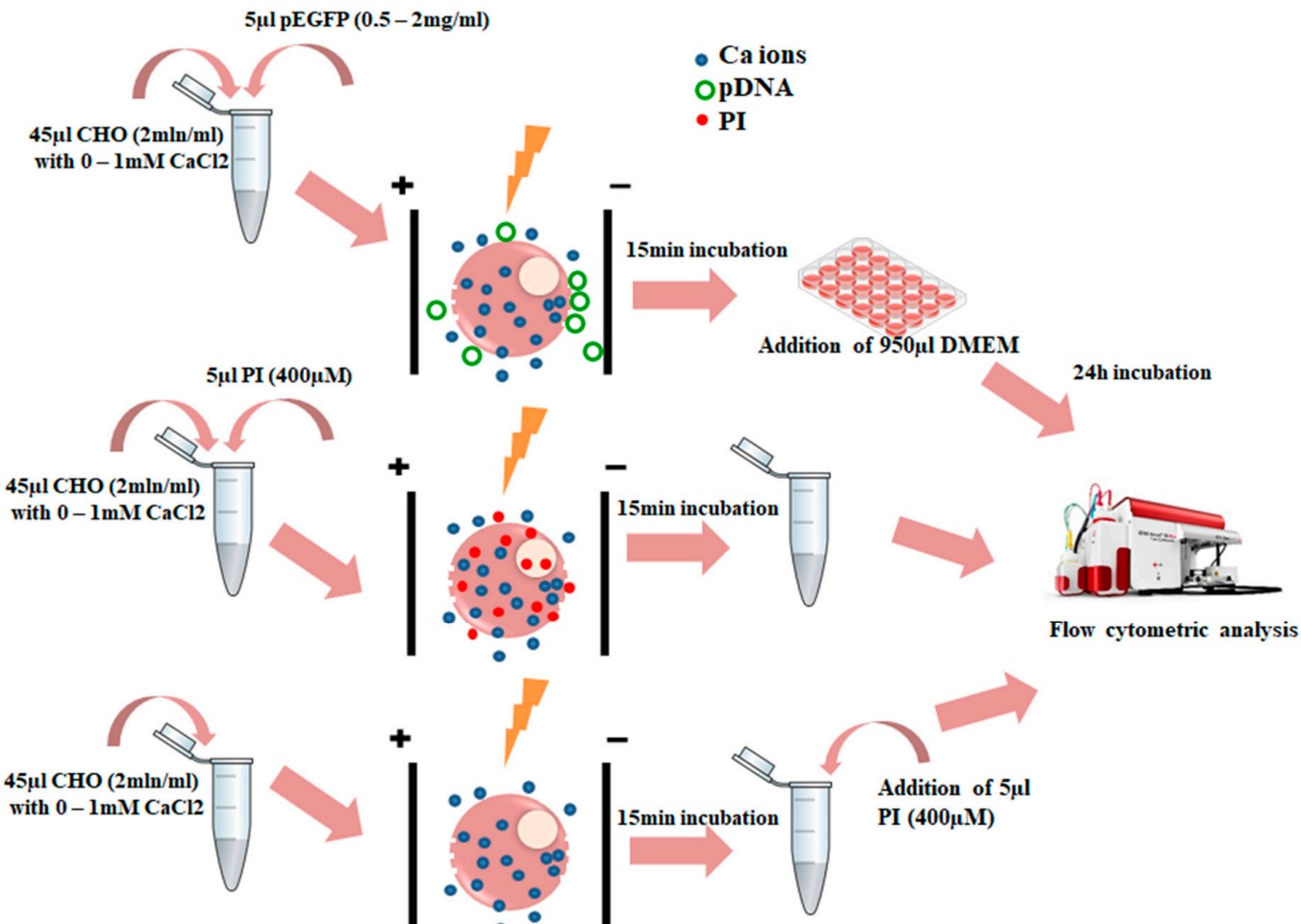

**Figure 1.** The principal scheme of the experiments performed in this study in order to evaluate pDNA and PI electrotransfer dynamics at different concentrations of CaCl₂.

*2.4. Electrotransfer of pDNA, Electrotransfer of Propidium Iodide (PI), and the Evaluation of Cell Viability*

For each experimental point, 45 µL of cell suspension ($9 \times 10^4$ cells) already supplemented with a different CaCl₂ concentration was mixed with 5 µL of the plasmid DNA solution to the final concentrations of 50, 100, 150, and 200 µg/mL pDNA in order to obtain a high electrotransfer efficiency with a minor effect on the cell viability [21]. After electroporation, the cells were placed in 24-well plates (TPP, Trasadingen, Switzerlands) for 15 min for recovery. After the recovery time, each well was supplemented with 950 µL of DMEM. Then, 24 h after electroporation, the growth medium was removed and cells were washed using 1× PBS and resuspended using 1× TryplE Express (Thermo Fisher Scientific, Washington, DC, USA). After subsequent incubation for 10 min at 37 °C, the cells were subjected to analyses of flow cytometry and determination of gene electrotransfer (GET) efficiency, mean fluorescence intensity (MFI), and cell viability. GET was evaluated as the percentage of the viable GFP-positive cells in the population, and the MFI (mean fluorescence intensity) was analyzed in the fraction of GFP-positive cells. Both GET and MFI were then used in the calculation of the total fluorescence intensity:

$$\text{Total fluorescence intensity} = \text{GET} * \text{MFI} \tag{1}$$

For the evaluation of the cell viability, a flow cytometry assay (FCA) was used. The number of cells to collect was set to be constant at $1 \times 10^4$ for all experimental points. Therefore, changes in the cell viability for each analyzed sample were reflected by changes in the

volume required to collect this constant number of cells. The values for the control samples were normalized to 100%, and the cell viability in other samples was calculated accordingly.

The overall GET was calculated by normalizing the transfection efficiency according to the number of counted viable cells:

$$\text{Overall GET} = \frac{\text{GET (\%)} * \text{Viability (\%)}}{100\%} \tag{2}$$

In order to distinguish cells according to the cell death pattern, the total decrease in cell viability (Cell death$_{pDNA+CaEP}$) was categorized as the calcium-electroporation-induced cell death (Cell death$_{CaEP}$) and pDNA-induced cell death (Cell death$_{pDNA}$) using the following equations:

$$\text{Cell death}_{pDNA+CaEP} = \text{Viability}_{Control}(\%) - \text{Viability}_{pDNA+CaEP}(\%); \tag{3}$$

$$\text{Cell death}_{CaEP} = \text{Viability}_{Control}(\%) - \text{Viability}_{CaEP}(\%) \tag{4}$$

$$\text{Cell death}_{pDNA} = \text{Cell death}_{pDNA+CaEP}(\%) - \text{Cell death}_{CaEP}(\%) \tag{5}$$

To evaluate the pore resealing after cell electroporation in the presence of $CaCl_2$, a propidium iodide (PI) assay was performed. For these experiments, 5 μL of PI (400 μM) (Sigma-Aldrich, St. Louis, MO, USA) was added to 45 μL of the cell mixture either before the delivery of the electric pulses (to evaluate the number of electroporated cells) or 15 min after the delivery of the electric pulses (to evaluate whether cells electroporated in the presence of $CaCl_2$ were able to reseal their pores).

### 2.5. Flow Cytometry

A Flow cytometric analysis was performed using a BD Accuri C6 (BD Biosciences, Franklin Lakes, NJ, USA) flow cytometer. For each experimental point, $1 \times 10^4$ cells were analyzed at a flow rate of (66 μL min$^{-1}$). For the evaluation of the pDNA electrotransfer efficiency, an FL-1 emission filter was used (excitation: 488 nm; emission: 533/30), and PI transfer dynamics were evaluated using an FL-2 emission filter (excitation: 488 nm; emission: 585/40 nm). All experiments and measurements were performed at room temperature; i.e., 22–25 °C.

### 2.6. Agarose Gel Electrophoresis

For the evaluation of the influence of $CaCl_2$ on the electrophoretic mobility of the pDNA molecules, 5 μL of plasmid was mixed with 45 μL of HEPES medium supplemented with different concentrations (0, 0.25, 0.5, or 1 mM) of $CaCl_2$. Then, the samples were loaded into 0.5% agarose gel. For the gel preparation, agarose (Thermo Fisher Scientific, Washington, DC, USA) was melted in 1× TAE buffer (Thermo Fisher Scientific, Washington, DC, USA) and mixed with ethidium bromide (Sigma-Aldrich, St. Louis, MO, USA) at a final concentration of 1 μg/mL. Each sample was mixed with 2–3 μL of 2× DNA loading dye (Thermo Fisher Scientific, Washington, DC, USA). Finally, a 1 kb DNA ladder (Thermo Fisher Scientific, Washington, DC, USA) was added to the gel, and the reaction was performed at 80 V for 45 min. For the visualization of the results, a UV transilluminator (Cleaver Scientific, Rugby, UK) was used.

### 2.7. Statistical Analysis

The results are presented as the mean ± standard error of the mean (SEM) of the datapoints obtained by performing at least three separate experiments. The statistical analysis was performed using Microsoft Excel; the graphs were produced using 12.5 GraphPad Prism 8 software. A two-tailed Student's *t*-test for the comparison of independent samples was used in order to assess the statistical significance of the differences between the groups. A *p*-value of <0.05 was considered to indicate statistical significance.

## 3. Results

The first part of the research was designed to evaluate whether low (0–1 mM) concentrations of $CaCl_2$ in the electroporation medium altered the efficacy of electrotransfection. The experiments were performed using two sets of pre-optimized electrical parameters and an optimal (high GET and high cell viability) pDNA concentration. The results, which are presented in Figure 2A, indicated that at 0 mM $CaCl_2$, a higher DNA transfection efficiency was obtained at a lower electric field strength (1200 V/cm) and a longer pulse duration (250 µs) as compared with a higher electric field strength (1400 V/cm) and a shorter (100 µs) pulse [11,22]. Interestingly, the pre-administration of $CaCl_2$ to the electroporation medium resulted in an inversely proportional dependence between the increment in the $CaCl_2$ concentration and the transfection efficiency (Figure 2A) as well as between the $CaCl_2$ concentration and the total fluorescence intensity (Figure 2B). Even low $CaCl_2$ concentrations (0.1 and 0.25 mM) in the electroporation medium induced a gradually decreasing GET and total fluorescence intensity in both electroporation protocols. The first relevant effect ($p < 0.001$) was observed when the $CaCl_2$ concentration was increased to 0.25 mM. In this case, the transfection efficiency decreased from 53.4% (0 mM) to 22.8% (0.25 mM) when an electric field of 1200 V/cm was applied and from 36.4% (0 mM) to 14.03% (0.25 mM) when a single pulse of 1400 V/cm was used. Further increases in $CaCl_2$ to 0.5 and 1 mM resulted in a drastic decrease ($p < 0.001$) in the number of GFP-positive cells as well as in the total fluorescence intensity.

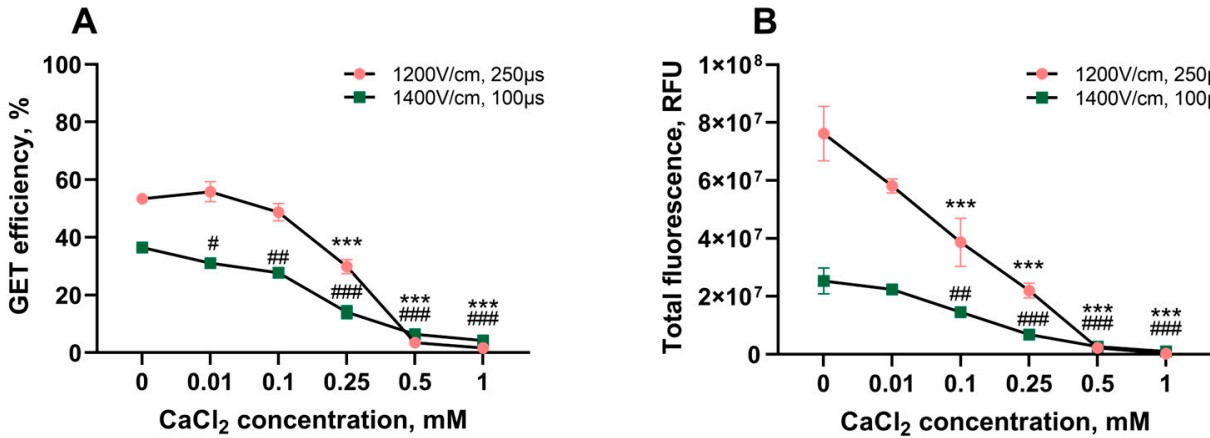

**Figure 2.** Dependence of pDNA electrotransfection efficiency (**A**) and total fluorescence intensity (**B**) on the strength of the applied electric field at different $CaCl_2$ concentrations. Cells were treated using one square pulse of 1200 V/cm for 250 µs or 1400 V/cm for 100 µs. Results were evaluated 24 h after electroporation. Error bars represent the standard error of the mean (SEM) of $n$ = 9 experimental replicates. In case standard errors were lower than actual data points, error bars are not visible. The statistical significance between the control (0 mM) and the 0.01, 0.1, 0.25, 0.5, and 1 mM $CaCl_2$ concentrations is denoted by *** (1200 V/cm) and #, ##, and ### (1400V/cm). # represents $p < 0.05$, ##—$p < 0.01$, and *** or ### represents the $p < 0.001$.

Since the results suggested that GET was significantly affected by $CaCl_2$, in the next part of the study we evaluated the influence of $CaCl_2$ on the cell viability. The data showed that calcium electroporation alone significantly reduced the cell viability when the $CaCl_2$ concentration exceeded 0.5 mM and an electric field of 1200 V/cm was used (Figure 3A). Even though such a significant decrease was not observed in the application of a 1400 V/cm strength pulse (Figure 3B), the simultaneous electrotransfer of pDNA and calcium revealed that a common additional cytotoxic effect occurred in a concentration-dependent manner for both electroporation protocols. Although very low concentrations of $CaCl_2$ (0.01–0.1 mM) in the electroporation medium induced only a slight decrease in the cell viability, the highest decrease was observed at the 0.25 mM concentration of $CaCl_2$: the viability decreased from 66.4 to 28.8% (1200 V/cm) and from 79.7 to 22.2% (1400 V/cm).

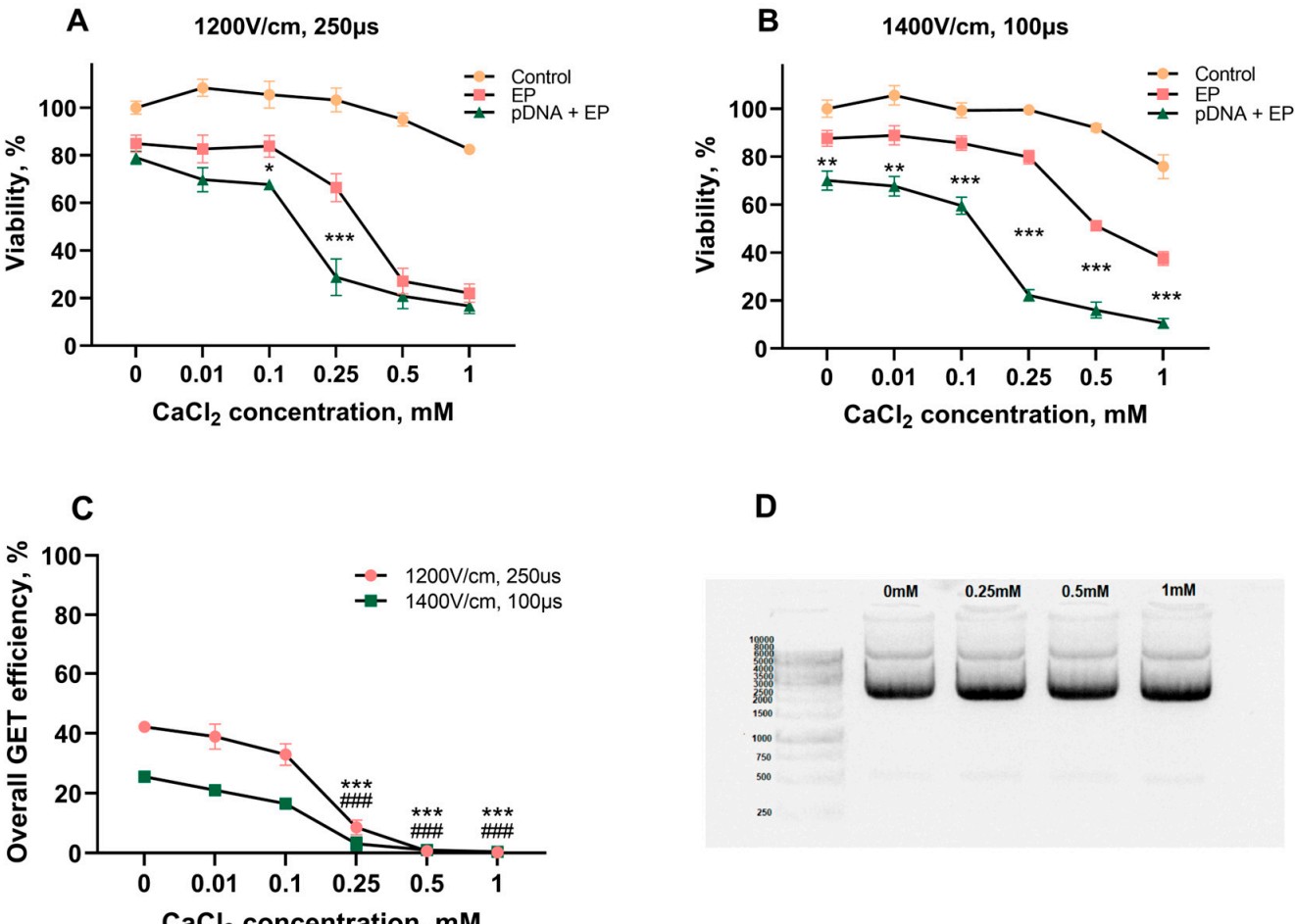

**Figure 3.** The effect of different CaCl$_2$ concentrations on cell viability (**A**,**B**), overall GET (**C**), and pDNA electrophoretic mobility (**D**). For panels A, B, and C, cells were treated using one square pulse of 1200 V/cm for 250 μs or 1400 V/cm for 100 μs. Results were evaluated 24 h after electroporation. Error bars represent the standard error of the mean (SEM) of $n$ = 9 experimental replicates. In case standard errors were lower than actual data points, error bars are not visible. The statistical differences between calcium electroporation alone (pink squares) and in combination with pDNA (green triangles) at different CaCl$_2$ concentrations are denoted by *, **, and *** (Panels A and B). For panel C, the statistical significance between the control (0 mM) and the 0.1, 0.01, 0.25, 0.5, and 1 mM CaCl$_2$ concentrations is denoted by *** (1200 V/cm) and ### (1400 V/cm). * represents the $p < 0.005$, **—$p < 0.01$ and *** or ### represents $p < 0.001$. For panel D, pDNA was electroporated using a single 1200 V/cm, 250 μs electric pulse; and electrophoretic mobility was evaluated via agarose gel electrophoresis.

In order to check whether the observed reduction in GET was the result of cell death, we normalized the transfection efficiency to the number of viable cells (Figure 3C). The obtained results revealed that when the CaCl$_2$ concentration exceeded ≥0.25 mM, the cell transfection rates were reduced significantly even though cells were still viable. With 0.25 mM CaCl$_2$ and a viability of 28.8%, only 8.6% of cells were GFP-positive when the electric field of 1200 V/cm was applied. A similar tendency appeared with the electric field of 1400 V/cm: only 3.1% of cells were transfected, and viability was still above 20%. A further increase in the CaCl$_2$ concentration to 0.5–1 mM further confirmed these observations and revealed that even though ~20% of cells were still viable, no transfection was detected for either electroporation protocol. These results suggested that the reduced GET efficiency was not solely the result of cell death, and that secondary mechanisms were implicated to some extent.



Since $Ca^{2+}$ can bind to DNA [23], we decided to investigate whether the observed phenomenon was the result of the altered electrophoretic mobility of pDNA; e.g., due to pDNA aggregation. For this experiment, the plasmid was mixed with electroporation medium containing different concentrations of $CaCl_2$, and agarose gel electrophoresis subsequently was performed. As presented in Figure 3D, there was no significant relationship observed between the pDNA mobility and the $Ca^{2+}$ concentration in the sample.

As shown previously, pDNA electrotransfection induced a significant cytotoxic effect on the cell viability when $CaCl_2$ was present before electroporation. In order to better understand the $Ca^{2+}$-induced decrease in transfection efficiency, cell death induced by calcium electroporation alone and by pDNA electrotransfer alone (Figure 4A,B) were evaluated separately. In addition to previous observations, the data revealed that the number of cells that had died due to the impact of pDNA increased with an increasing $CaCl_2$ concentration. From 5.9% at 0 mM $CaCl_2$ in the medium, the proportion of pDNA-related cell deaths increased to 37.6% at 0.25 mM $CaCl_2$ when an electric field of 1200 V/cm was used ($p < 0.001$). A similar tendency was observed when an electric field of 1400 V/cm was applied: the pDNA-related cell death increased from 17.6% (0 mM) to 56.9% (0.25 mM) ($p < 0.001$) Since electroporation alone—with 0.5–1 mM $CaCl_2$ pre-administered to the medium—was responsible for a significant decrease in the cell viability, the effect observed previously was no longer detected.

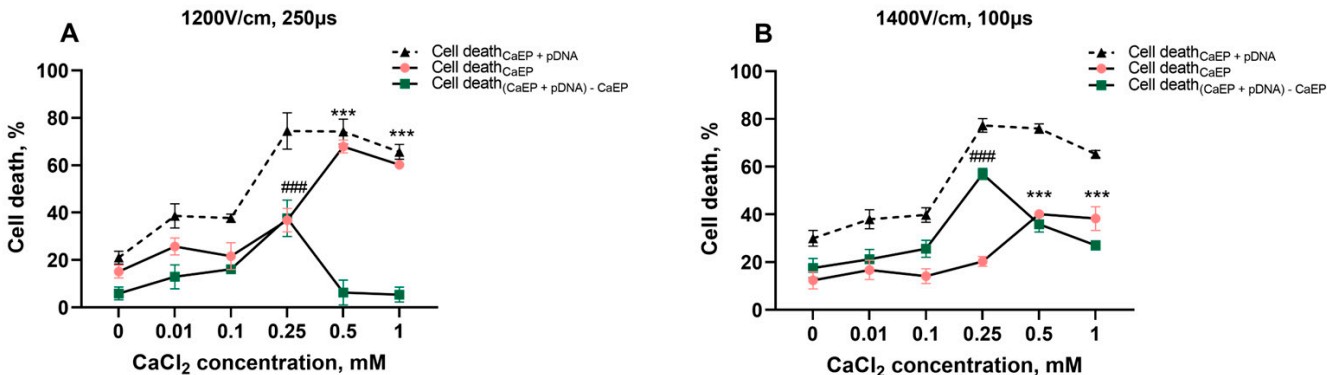

**Figure 4.** The dependence of cell death on the applied electric field at different $CaCl_2$ concentrations. Cells were treated using one square pulse of 1200 V/cm for 250 μs (**A**) or 1400 V/cm for 100 μs (**B**). Results were evaluated 24 h after electroporation. Error bars represent the standard error of the mean (SEM) of $n = 9$ experimental replicates. In case standard errors were lower than actual data points, error bars are not visible. The statistical significance between the control (0 mM) and the 0.1, 0.01, 0.25, 0.5, and 1 mM $CaCl_2$ concentrations is denoted by *** (cell death induced by calcium electroporation) and ### (cell death induced by pDNA). *** and ### represents the $p < 0.001$.

In order to gain a deeper understanding of the intracellular processes related to $Ca^{2+}$ electroporation, a PI assay was performed. PI was added at two different time points: before or 15 min after pulse application with 0–1 mM $CaCl_2$ present in the medium. As illustrated in Figure 5A, the increase in calcium concentration was related to the decrease in the percentage of PI-positive cells. This phenomenon indicated that calcium could enhance the rate of pore resealing, which we previously demonstrated [24,25]. The number of PI-positive cells obtained 15 min after pulse application revealed that membrane pores had already resealed within this period for the full range of $CaCl_2$ concentrations tested (Figure 5B). Even though the integrity of the cell membranes was restored, the results (Figure 3B,C) proposed that the cell viability was strongly affected within 24 h after calcium electroporation, which suggested that cell death mechanisms—independent from resealing of pores—were triggered as a response to the treatment with $Ca^{2+}$.

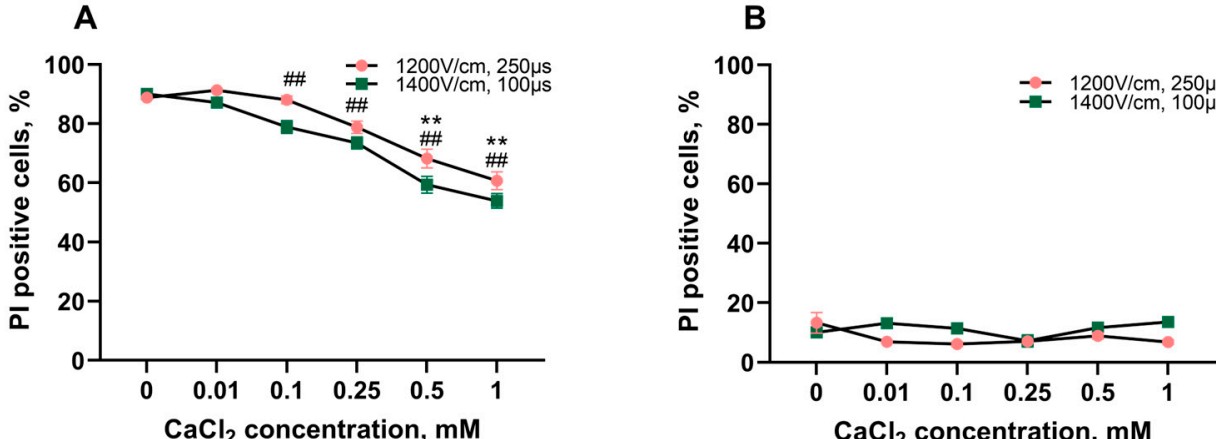

**Figure 5.** The dependence of PI electrotransfer efficiency on the strength of the applied electric field at different CaCl$_2$ concentrations either before (**A**) or after (**B**) electroporation. Cells were treated using one square pulse of 1200 V/cm for 250 μs or 1400 V/cm for 100 μs. Error bars represent the standard error of the mean (SEM) of $n = 9$ experimental replicates. In case standard errors were lower than actual data points, error bars are not visible. The statistical significance between the control (0 mM) and the 0.1, 0.01, 0.25, 0.5, and 1 mM CaCl$_2$ concentrations are denoted by ** (1200 V/cm) and ## (1400 V/cm). ** and ## represents the $p < 0.01$.

We also evaluated the importance of different plasmid DNA concentrations for the decrease in the efficiency of DNA electrotransfer induced by Ca$^{2+}$. As already described, electroporation of 1200 V/cm for 250 μs resulted in a significantly higher electrotransfer efficiency, and therefore the experiments were performed using these electric field parameters (Figure 6). The increase in pDNA concentration resulted in a significant increase in the transfection efficiency (at 0 mM CaCl$_2$) (Figure 6A). A similar tendency was observed when the CaCl$_2$ concentration was increased to 0.01 mM and 0.1 mM. Conversely, when the CaCl$_2$ concentration was further increased to 0.25 mM, an increase in the pDNA concentration did not further result in an increase in the transfection efficiency. The administration of a 0.01 mM CaCl$_2$ concentration to the electroporation medium resulted in an increase in the transfection efficiency from 24.3% (50 μg/mL) to 61.1% (200 μg/mL), whereas only an insignificant increase in the transfection efficiency was observed when the electroporation was performed at the 0.25 mM concentration of CaCl$_2$. Again, these results implied that 0.25 mM CaCl$_2$ is a threshold value in the process of pDNA electrotransfection and is important to the changes in cell viability. The successive increase in the CaCl$_2$ concentration supported this hypothesis, and no further increase in the transfection efficiency was observed when the CaCl$_2$ concentration in the medium was $\geq$0.5 mM. As expected, the evaluation of the total fluorescence intensity at different pDNA concentrations revealed an increase in fluorescence in accordance with the increasing plasmid concentration (Figure 6B). In addition, in agreement with previous results, the increase in the CaCl$_2$ concentration resulted in a gradual decrease in the total fluorescence for all pDNA concentrations used. The evaluation of the cell death (Figure 6C) indicated that the cell death occurring during Ca$^{2+}$ electroporation was dependent on the pDNA concentration. Nevertheless, the results also confirmed previous observations: when the CaCl$_2$ concentration exceeded 0.25 mM, the combination of Ca$^{2+}$ and pDNA significantly affected the cell viability. In this case, the viability decreased from 66.4% (electroporation alone) to 44.5, 28.8, 25.5, and 23.8% when plasmid concentrations of 50, 100, 150, and 200 μg/mL were used, respectively. However, only a slight decrease in the cell viability was observed when lower CaCl$_2$ (0.01 and 0.1 mM) concentrations and various concentrations of pDNA were used. It is worth noting that calcium electroporation (0.5 and 1 mM) alone induced a decrease in the cell viability to 27 and 22%, respectively. Surprisingly, a further increase in the plasmid concentration had no or only minimal effects on an additional decrease in the cell viability. Normal-

ization of the transfection efficiency according to the cell viability (Figure 6D) confirmed previous observations that $CaCl_2$ reduced the transfection efficiency of plasmid DNA in a concentration-dependent manner. In accordance with previously presented results, only 10.2% (50 μg/mL), 8.6% (100 μg/mL), 6.5% (150 μg/mL), and 5.9% (200 μg/mL) of GFP-positive cells were detected with 0.25 mM of $CaCl_2$ in the electroporation medium 24 h after electroporation.

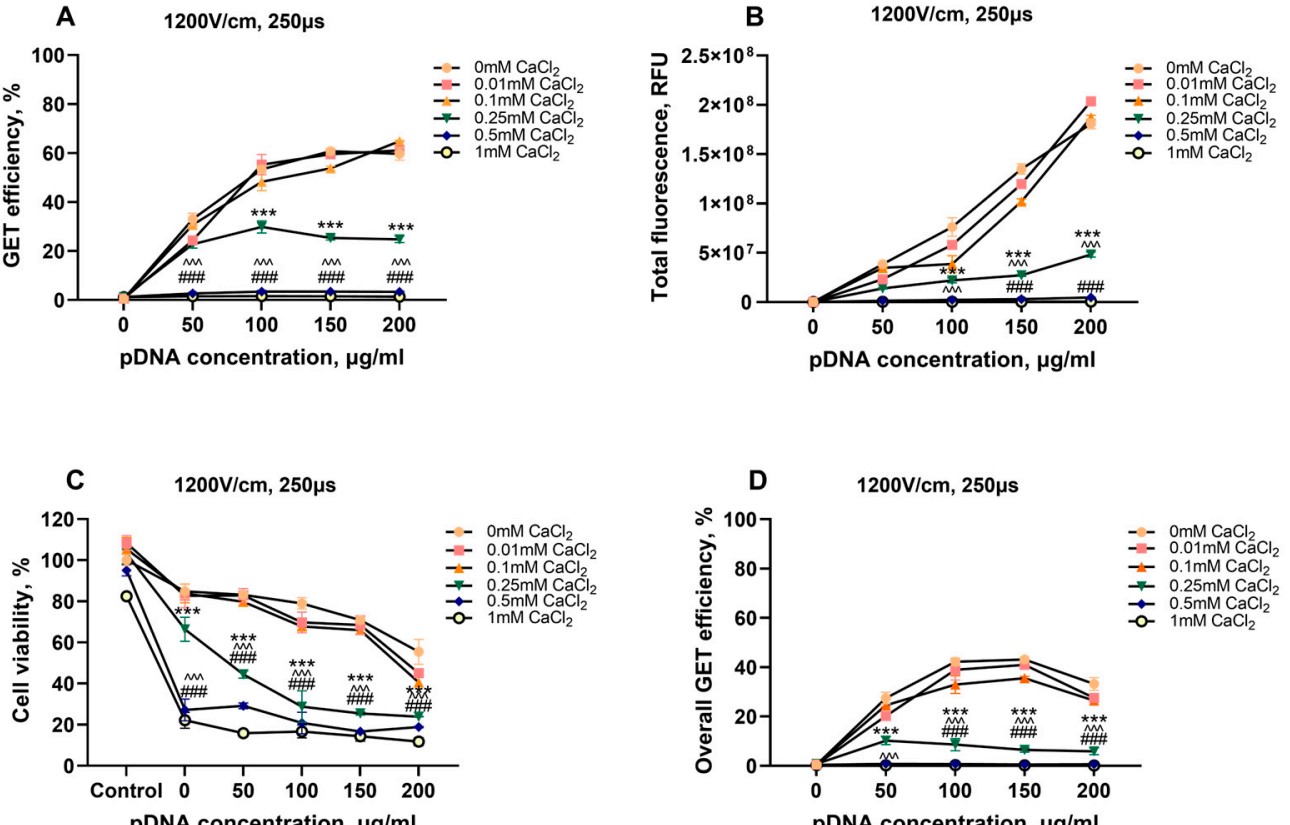

**Figure 6.** The dependence of pDNA electrotransfer efficiency (**A**), GFP total fluorescence intensity (**B**), cell viability (**C**), and normalized electrotransfer efficiency (**D**) on pDNA and $CaCl_2$ concentrations in the electroporation medium. Cells were treated using one square pulse of 1200 V/cm for 250 μs. Results were evaluated 24 h after electroporation. Error bars represent the standard error of the mean (SEM) of *n* = 9 experimental replicates. In case standard errors were lower than actual data points, error bars are not visible. The statistical differences between the control (0 mM) and the 0.25, 0.5, and 1 mM $CaCl_2$ at different pDNA concentrations used are denoted by ***, ###, and ^^^, respectively; three symbols denote $p < 0.001$.

## 4. Discussion

$Ca^{2+}$ is one of the most prevalent and important compounds in both intracellular and extracellular environments. It is estimated that the intracellular concentration of $Ca^{2+}$ is in the micromolar range (0.1–0.2 μM), whereas extracellular calcium concentration varies between 1 and 3 mM [26]. Therefore, it is not surprising that $Ca^{2+}$ influences various processes such as gene electrotransfer [20,24,25,27–29]. The inhibitory effect of $Ca^{2+}$ on electrotransfer efficiency and luciferase fluorescence intensity has been observed in studies conducted in vivo while using relatively high $CaCl_2$ concentrations that varied between 10 mM and 1 M [20] However, when considering the pronounced difference in intra- and extracellular concentration, it is possible that even small alterations in the amount of $Ca^{2+}$ may induce significant impacts on diverse cellular processes.

In the current study, we investigated the impact of low (0.01–1 mM) concentrations of $CaCl_2$ on the efficiency of pDNA electrotransfer using several electroporation protocols. The

obtained results clearly demonstrated a decreased transfection efficiency with an increasing $CaCl_2$ concentration. The results revealed that a low $CaCl_2$ concentration of 0.25 mM pre-administered to the electroporation medium reduced the electrotransfer efficiency of the pDNA molecules by ~1.5-fold (Figure 2A,B) and altered the cell viability in a concentration-dependent manner (Figure 3A,B). The data also suggested that the decrease in GET was not solely the result of a decrease in the number of viable cells. The results demonstrated that a higher $CaCl_2$ concentration was related to a lower fraction of GFP-positive cells being present in the viable population. When the $CaCl_2$ concentration was >0.25 mM, the percentage of GFP-positive cells in the viable population remained similar (~20%), but a gradual decrease in GET was still observed; this suggested the influence of secondary $Ca^{2+}$-related mechanisms that affected GET. In this context, we speculated that gradual $Ca^{2+}$-related energy depletion can affect many endocytotic, pDNA intracellular migration, transcription, and translation pathways needed for cell electrotransfection.

The observed phenomenon led to two hypothesized mechanisms. One is that calcium modifies pDNA properties that are essential for efficient electrotransfer, thereby resulting in a reduced efficiency of the electrotransfer of pDNA. This hypothesis arose from studies that showed the neutralization of DNA charge induced by $Ca^{2+}$ and therefore reduced electrophoretic forces during electroporation [18,23] One of these studies also proposed that pDNA has binding sites for $Ca^{2+}$ and is even able to compete with calmodulin for the binding of calcium [23]. In order to test the latter assumption, we performed agarose gel electrophoresis (Figure 3D); however, no difference in pDNA electrophoretic mobility was observed within the range of $CaCl_2$ concentration (0–1 mM) tested. Even though the results suggested that pDNA electrophoretic mobility was not altered, other studies revealed that sub-millimolar levels of $Ca^{2+}$ can induce conformational changes in DNA molecules. Calcium ions can negatively affect not only electrotransfection efficiency [18] but also gene transcription and translation, and they can even entail epigenetic mechanisms that can significantly alter gene expression [30,31]. These findings may explain the main result obtained in our study: a $Ca^{2+}$-induced decrease in GFP-encoded plasmid electrotransfer efficiency.

Even though our results revealed that cell death was not solely responsible for the decreased level of GET, an observed significant cytotoxic effect of the $Ca^{2+}$ and pDNA combination could not be discriminated. Our data proposed that the increasing $CaCl_2$ was related to an increasing rate of pDNA-induced cell death and showed a maximum effect at the 0.25 mM concentration of $Ca^{2+}$ (Figure 4A,B). This allowed us to consider the other possible hypothesis: an increased potency of pDNA electrotransfer associated with the presence of calcium that could lead to an increase in the rate of cell death. This hypothesis was adopted from a study published by Neumann et al. [32], who investigated $Ca^{2+}$-mediated DNA adsorption to yeast cells. Subsequent studies have also suggested that $Ca^{2+}$ may operate as a bridge and reduce electrostatic repulsion between the negatively charged lipid bilayer and pDNA [33]. Furthermore, computer simulations have revealed that calcium ions can stabilize DNA–membrane complexes [34]. Theoretically, more stable DNA–lipid complexes are formed, more DNA molecules enter the cell, and the transfection efficiency is higher. However, another study [33] showed that even though $Mg^{2+}$ induced the increased adsorption of pDNA on the cell membrane, the resultant efficiency of gene electrotransfer was decreased. The authors suggested that fewer DNA molecules could enter the cell due to strong $Mg^{2+}$-induced interaction between DNA and the lipid bilayer. Assuming this was also the case with $Ca^{2+}$, pDNA would not be able to translocate through the membrane after DNA–membrane complex formation, and in addition, the pores would not be able to reseal normally, which potentially would lead to higher rates of cell death (as we observed in our study).

Calcium electroporation alone was tested as a novel anticancer treatment both in vitro and in vivo and showed a significant decrease in cell viability [35–41]. Our results also revealed a dramatic decrease in the cell viability when higher concentrations (0.5–1 mM) of $CaCl_2$ were used (Figure 4A,B). Successively, we decided to evaluate the importance of

$Ca^{2+}$ to the dynamics of pore resealing (Figure 5A,B). Experiments were performed with no pDNA in the sample because PI intercalated with pDNA before electroporation and it was impossible to distinguish the signal that arose from PI interaction with pDNA and its interaction with genomic DNA. In agreement with our previous study [24], the results revealed an increase in the rate of pore resealing with an increasing $Ca^{2+}$ concentration, and the measurement of PI-positive cells 15 min after the incubation revealed that the pores had resealed for all the $CaCl_2$ concentrations applied. This observation was also in concordance with our previous study [42], which demonstrated that secondary processes took place after electroporation and that pore resealing was only one of the factors that affected the cell viability. Indeed, calcium electroporation might induce ATP depletion leading to pronounced cell death and subsequent disturbances in natural processes such as gene expression, metabolism, and proliferation [43,44]. This may explain why higher (0.5–1 mM) $CaCl_2$ concentrations failed to reduce the efficiency of pDNA transfection.

In general, higher concentrations of pDNA in cell suspension lead to a higher transfection efficiency and expression as more DNA molecules enter the cell. Our results confirmed this assumption by showing a significant increase in the transfection efficiency with an increasing pDNA concentration when $Ca^{2+}$ was not present in the electroporation medium (Figure 6A,B). The latter effect disappeared when the calcium concentration exceeded 0.25 mM and the increase in pDNA concentration no longer resulted in an increase in GFP-positive cells. Interestingly, even when the transfection efficiency was within the plateau, a decrease in the cell viability with an increasing pDNA concentration in the sample was still observed (Figure 6C). This implied that observed decrease in cell viability was associated not only with absorption of DNA on the cell membrane but also with other factors such as cytosolic DNA sensors. This hypothesis arose from studies that showed DNA intracellular transfer to be capable of activating cytosolic DNA sensors and therefore led to the upregulation of cytokines and chemokines and initiated the mechanisms of cell death [45]. As discussed previously, we suggest that part of the DNA forms a strong bond with the plasma membrane. However, a higher pDNA concentration results in an increase in the total number of DNA molecules, which might still lead to a higher amount of DNA molecules entering the cell and therefore the activation of various cytosolic sensors.

## 5. Conclusions

Altogether, the results presented in the current study indicated that even a very low amount ($\leq 1$ mM) of calcium was sufficient to impose negative effects on the efficiency of DNA transfection with a threshold value of a 0.25 mM concentration. If this threshold was reached, the pDNA concentration had no influence on the transfection efficiency, and the cell viability was still diminished significantly. These observations suggested that intracellular delivery of $Ca^{2+}$ can serve and be successfully exploited as a tool for controlling GET. Even though our study defined the general guidelines for the application of $Ca^{2+}$ in tandem with pDNA in vitro, we believe that the obtained data on the interdependencies of $Ca^2$ and transfection efficiency might provide valuable insights into gene electrotransfer in vivo to reach the expected outcome.

**Author Contributions:** Conceptualization, R.P. and S.Š.; methodology, B.J.; software, P.R.; validation, R.P., S.Š. and P.R.; formal analysis, B.J.; investigation, R.P.; resources, M.M.; data curation, M.M.; writing—original draft preparation, R.P.; writing—review and editing, M.M. and S.Š.; visualization, R.P.; supervision, S.Š.; project administration, S.Š.; funding acquisition, S.Š. All authors have read and agreed to the published version of the manuscript.

**Funding:** This research was funded by the European Social Fund according to the activity 'Improvement of researchers' qualification by implementing world-class R&D projects' of Measure No. 09.3.3-LMT-K-712-01-0188.

**Institutional Review Board Statement:** Not applicable.

**Informed Consent Statement:** Not applicable.

**Data Availability Statement:** Data sharing is not applicable.

**Conflicts of Interest:** The authors declare no conflict of interest.

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
