# Peer review of "The Influence of Calcium Ions on the Electrotransfer Efficiency of Plasmid DNA and Cell Viability"

_applsci, doi:10.3390/app13031983_

Round 1

Reviewer 1 Report

The Authors presented a valuable study concerning the influence of calcium ions on the electrotransfer efficiency of pDNA and cell viability. The paper is well written and presented; however, there are some comments to be addressed:

- line 89 - what is sterile water? is it MilliQ?

- justify the selection of used parameters:  1400V/cm, 100μs and 1200V/cm, 250μs; why not the same pulse duration, or the same voltage?

- what assay was used for cell viability? it should be clarified and differentiated the usage of PI to permeabilization studies and viability. 

- in wat temperature were cells analyzed by flow cytometry?

- the Authors demonstrated decreasing GET efficacy with the increased calcium concentration and simultaneously decreasing cell viability. Do you suppose that if cells are more affected by calcium electroporation, their ability to uptake pDNA is weaker? 

- fig 4B, missing line labels in legend 

- fig.5B is not necessary; it is rather evident that PI will be not delivered afer minutes post electroporation because of membrane resealing. In the description (lines 220-225) you can not compare this result to results measured after 24h (cell viability).  This part should be removed or revised. 

Reviewer 2 Report

Manuscript reports on the potential influence of calcium ions on gene deliver using electrotransfer. Authors present some interesting aspects to the concept. There are issues that the authors should resolve to improve this manuscript.

In the Introduction, the authors state that cell electrotransfecton is relatively low efficiency. Two points about this statement, while taken at face value one could argue that there is some validity to this statement, however, particularly for in vitro delivery, which is the focus of this report, it is not as factual as it once was. It is possible to get very high efficiency in vitro, dependent on delivery parameters and cell lines used.  In addition, the references used were not the best to validate this statement.

In Methods and Results section, the authors used very high concentrations of plasmid and it is not clear why these levels were chosen. This should be clarified in the manuscript.

In the first part of the Results section the authors mention the "importance of electrophorectic forces", this is an unqualified statement as 250 us pulses is not typically associated with the notion of electrophorectic forces in relation to electrotransfer. these are typically much longer in the 10s or 100s of ms.

In the results section in the paragraphs relating to figures 3 and 4. These sections are quite confusing in how the results are described and explained. Part of the issue is the poor English and part is the use of terms that are very similar. There is too much overlap in the terms and it is not clear how to separate each piece of data particularly since to obtain some data there need to be combining of factors.

Sentence on lines 280-282 is confusing and not structured properly.

In the Discussion section, the authors on lines 282-284 mention a reason relating efficiency and viability. However, because the cells and pDNA are in solution there is also an issue with contact between the two components and with less viable cells it is possible that there is less potential for interaction leading to lower level of transfected cells which would be related to viability. This is much more likely than what the Authors propose.

Statement on lines 285-286 does not seem to correlate with the actual data. Either is poorly written or the data is misinterpreted.

The manuscript needs a major edit. There are many grammatical errors and misuse of words making the manuscript difficult to read.

Reviewer 3 Report

Detailed comments are attached.

Round 2

Reviewer 3 Report

The manuscript reads much better, but some minor formatting changes are required for the equations. I give an example for Equation (1) in the attached file. 

Author Response

All reviewer comments were addressed. See the main text.